# Micromotor Manipulation Using Ultrasonic Active Traveling Waves

**DOI:** 10.3390/mi12020192

**Published:** 2021-02-13

**Authors:** Hiep Xuan Cao, Daewon Jung, Han-Sol Lee, Gwangjun Go, Minghui Nan, Eunpyo Choi, Chang-Sei Kim, Jong-Oh Park, Byungjeon Kang

**Affiliations:** 1Korea Institute of Medical Microrobotics, Gwangju 506813, Korea; hiep.caoxuan@kimiro.re.kr (H.X.C.); jungdaewon@kimiro.re.kr (D.J.); hansol3607@gmail.com (H.-S.L.); gwangjun124@kimiro.re.kr (G.G.); myeonghea94@kimiro.re.kr (M.N.); eunpyochoi@jnu.ac.kr (E.C.); 2School of Mechanical Engineering, Chonnam National University, Gwangju 61186, Korea; 3Robotics Engineering Convergence, Chonnam National University, Gwangju 61186, Korea

**Keywords:** acoustic manipulation, active traveling wave, ultrasonic actuation, non-contact manipulation, particle manipulation, targeted drug delivery

## Abstract

The ability to manipulate therapeutic agents in fluids is of interest to improve the efficiency of targeted drug delivery. Ultrasonic manipulation has great potential in the field of therapeutic applications as it can trap and manipulate micro-scale objects. Recently, several methods of ultrasonic manipulation have been studied through standing wave, traveling wave, and acoustic streaming. Among them, the traveling wave based ultrasonic manipulation is showing more advantage for in vivo environments. In this paper, we present a novel ultrasonic transducer (UT) array with a hemispherical arrangement that generates active traveling waves with phase modulation to manipulate a micromotor in water. The feasibility of the method could be demonstrated by in vitro and ex vivo experiments conducted using a UT array with 16 transducers operating at 1 MHz. The phase of each transducer was controlled independently for generating a twin trap and manipulation of a micromotor in 3D space. This study shows that the ultrasonic manipulation device using active traveling waves is a versatile tool that can be used for precise manipulation of a micromotor inserted in a human body and targeted for drug delivery.

## 1. Introduction

Various approaches for non-contact manipulation of micromotors, which are propelled by chemical and physical stimuli, have been widely studied in the field of targeted drug delivery [1,2,3,4]. The physical stimuli have the merit of high controllability, thus numerous research groups demonstrate the manipulation by using physical sources, such as optical, magnetic, and acoustic fields [4]. Drinkwater’s group presented the acoustic method capable of micromotor manipulation in the water, whereas Nelson’s group has shown that bioinspired micromotor can be actuated by the magnetic field in vivo, and Liu’s group used the optical method for delivering the anticancer drug-loaded motors to the cancel cell [5,6,7].

Among these methods, magnetic and acoustic fields-based manipulations have recently attracted great attention for in vivo applications owing to their penetration capability in biological tissue [8,9,10,11].

In previous studies, magnetic energy was primarily considered as a powering source to manipulate micromotors in in vivo experiments, because the magnetic actuation method has the advantage of powerful force and torque [12,13,14]. However, it should be coated with magnetic materials on the micromotors, which require extra process and decrease the biocompatibility of the micromotor [15,16]. Moreover, magnetic energy is not able to generate a sink or source node that provides dragging or repelling force to the micromotors from a point. These features bring that low efficiency of the micromotors collection and targeting.

Contrastively, acoustic energy can generate the pressure nodes and manipulate nonmagnetic objects [17,18,19]. Thanks to the advantage of acoustic energy, several ultrasonic-based manipulation approaches are introduced in the previous studies. In these studies, a standing-wave generally used for generating acoustic radiation force in the parallel multidimensional transducer arrays [20,21,22]. However, this structure is not suitable for an in vivo environment, because the configuration with opposed transducer pair cannot be used in patients depending on the treatment area. The traveling wave, by contrast, uses a phase pattern controlling for the particle trapping and manipulation without reflectors, thus transducers can be placed in the single-sided plane [23,24,25]. This traveling wave uses passive and active control of acoustic phase pattern. The passive method uses the physical control method by using extra structures, like acoustic metamaterial and lens, thus it needs redesigned structures for changing the phase pattern. Because of the limitation, the passive system needs a motorized stage for real-time manipulation [6]. This motorized stage method can change the trapping point, but it has a problem in moving the point through a vertical axis into the inner body. In this aspect, the active method is a suitable solution that can change the individual phase values by the electronic control method. Using the active method, the trapping point is regenerated in the 3D space without any additional driving device.

The active traveling type uses several phase patterns, such as vortex, bottle, and twin trap, for particle manipulating [24,26,27]. Among them, the twin trap, which has the phase pattern of two cylindrical beams, is the suitable type that can show relatively strong acoustic trapping force with a wide range of steering angles [24]. This twin trap shape is created by the specific phase difference in the arranged acoustic transducers, and a hemispherical arrangement is used for the Ultrasonic Transducer (UT) array. The hemispherical arrangement can maximize the acoustic pressure, thus the intensity of acoustic radiation force is increased at the focal point.

Based on the designed UT array model, the expected trapping points are simulated in the finite element analysis software (COMSOL lnc., Burlington, MA, USA). The simulation result can show that the calculated phase values can generate desired trapping positions, or not. The overall performance is tested by manipulating the micromotor to follow the pre-programmed path, and a closed-loop control system is applied to improve the result. The purpose of this study is to present a novel 3D manipulation system for a 400-μm micromotor. This micromotor can be trapped and manipulated by only using electronic phase control, and the system is designed for the underwater condition. Because of these features, our system will be developed for an advanced targeted drug delivery technique in future work.

In this paper, the phase control mechanism, simulation methodology, and control system for the UT array model are presented in chapter 2. The experiment setup and micromotor manipulation results through in vitro and ex vivo experiments are referred to the following chapter 3. The conclusion of this study is in the final chapter, 4.

## 2. Materials and Methods

### 2.1. Physical Principles for Acoustic Radiation Force

The complex acoustic pressure $P_{j}$ at the point *r*, emitted from a single ultrasonic transducer, can be expressed as follows.
(1)$$P_{j}\left(r\right)=P_{0}A\frac{D_{f}\left(\theta \right)}{d}e^{i\left(\varphi +kd\right)}$$
where $P_{0}\;$is transducer amplitude power in constant, and A is the peak-to-peak amplitude of the excitation signal. $D_{f}\left(\theta \right)$ is a far-field directivity function that depends on the angle θ between the transducer normal and point r. Moreover, *d* is the propagation distance in free space, φ is the phase delay of the transducer, k=2π/λ is the wavenumber, and λ is the wavelength. The total acoustic field (P) generated by N transducers is the addition of the individual pressure that is $P=\overset{N}{\underset{j=1}{\sum }}P_{j}\left(r\right)$.

Under this pressure field, the acoustic radiation force applied to the Rayleigh micromotors (Radius of micromotors ≪ Wavelength of field) can be estimated by using the Gor’kov potential field [15,16,18]. The radiation force, F, and the potential filed, U, can be given by:(2)F=−∇U
(3)$$U=V_{P}\left[f_{1}\frac{1}{2}k_{0}\langle {\left|p_{1}\right|}^{2}\rangle -f_{2}\frac{3}{4}\rho_{0}\langle {\left|v_{1}\right|}^{2}\rangle \right]$$
(4)f1=1−kpk0 and f2=2(ρpρ0−1)2ρpρ0+1
where $\langle {\left|p_{1}\right|}^{2}\rangle$ and $\langle {\left|v_{1}\right|}^{2}\rangle$ are the mean squared values of the pressure at the trapped position and the particle’s velocity, while $V_{P}$ is the volume of the particle. $k_{0}$ and $\rho_{0}$ are compressibility and density of the host medium, and the subscript p means values for the particle. $f_{1}$ and $f_{2}$ are monopole and dipole coefficients.

### 2.2. Analysis of Ultrasonic Transducer Array Model

The physical focal point is decided by the radius of curvature in the UT array model and overlapped acoustic filed, which creates a single beam at the point. This focal position can be manipulated as shown in Figure 1a [27,28].

The first step of the manipulation mechanism is that calculating the number of acoustic wave cycles (*N*), which is defined as follows.
(5)$$Number\;of\;cycles\;\left(N\right)=\frac{\sqrt{{\left(P_{X}-T_{Xi}\right)}^{2}+{\left(P_{Y}-T_{Yi}\right)}^{2}+{\left(P_{Z}-T_{Zi}\right)}^{2}}}{\lambda }$$
where $\left(P_{X},\;P_{Y},\;P_{Z}\right)$ is the desired focal position, $\left(T_{Xi},\;T_{Yi},\;T_{Zi}\right)$ is individual transducer position in the array model, and λ is the wavelength of acoustic wave.

In the next step, the phased delay is calculated for each transducer according to the equation below:(6)$$Phase\;delay\;value\;\left(rad\right)=2\cdot \pi \cdot \left(1-f^{*}\left(N\right)\right)$$
where the function of $f^{*}\left(N\right)$ prints the decimal part of these cycle values, thus 1 − $f^{*}\left(N\right)$ means that the margin of a last one cycle at the focal point. Based on these values, homogeneous acoustic waves are overlapped at the desired focal point, and a single beam is reformed in the point.

The UT array is designed based on the hemispherical arrangement model with 16 identical transducers in this study as shown in Figure 1b. The transducers are divided into 2 parts from the left and right sides of the *Z*-axis, and the π-phase difference is applied to each side for making two cylindrical beams at the focal point, and these beams tweeze the micromotors. This trapping method is called a “twin trap”, and the focused model has a focal distance of 36.47 mm. The array model makes the space of overlapped acoustic waves as a sphere of 15 mm diameter, thus an inscribed cube (8.66 cubic mm) in the sphere is the theoretical workspace for the 3D manipulation. In theory, the trapped particle can be manipulated within the workspace, but when the focal point is regenerated far from the physical focal point, sidelobes are also created in the field. These lobes reduce the trapping force of a focal point, thus an optimal workspace has to be decided in the experiment section.

Based on the proposed UT array model, the acoustic radiation field can be simulated in COMSOL Multiphysics with a 1MHz frequency for 16 transducers. Figure 2 shows the normalized radiation fields and force values at the physical focal point of the array (0, 0, 0) and the manipulated point (1, 1, 1). In the simulation, the radiation force (nN) of each axis [$F_{x}$, $F_{y}$, $F_{z}$] are estimated as [2000, 280, 150] and [1600, 220, 100] at the physical focal point and manipulated point. In Figure 2b, the acoustic beam is manipulated into the (1, 1, 1) by using Equations (5) and (6). When the desired focal point $\left(P_{X},\;P_{Y},\;P_{Z}\right)$ is redefined as (1, 1, 1), each phase delay value is also calculated based on the modified number of cycles (N). These delay values make sure that the emitted waves from 16 transducers are re-overlapped at the new point. As shown in Figure 2b, the changed position of trapping nodes is presented, and these radiation force graphs show the typical feature of the twin trap that the vertical force ($F_{z}$) is the weakest force among each axis force.

### 2.3. Design of Control System

Each transducer of the UT array is operated with 1 MHz and the bipolar voltage amplitude of 60 Vpp, and the position of a micromotor is controlled using the phase modulation. Thus, the designed controller function can be express by the phase modulation function of each transducer, and the total phase of the $i^{th}$ transducer to manipulate a trapping point is defined as follows:(7)$$F_{ultrasound}=f\left(\varphi_{1},\;\varphi_{2},\;\cdots \varphi_{16}\right)$$
(8)$$\varphi_{i}=2\pi \left(1-f_{i}\left(N\right)\right)$$
where $\varphi_{i}$ is the phase delay in radian on the $i^{th}$ transducer. *N* and $f_{i}\left(N\right)$ is the number of acoustic wave cycles and the decimal part of the last cycle compare with one wavelength, respectively.

Figure 3 describes an overall block diagram of control hardware including a PC, FPGA module, vision system, a custom amplifier, power supply, and UT array. NI PCIe-7852R with FPGA module is hosted in the PC that is responsible for generating square wave with the voltage magnitude of 3.3 V, and two input signals are used to control bipolar voltage from the amplifier board. Namely, one signal controls the positive pole of the bipolar output voltage and the other one controls its negative pole. For amplifying the voltage from 3.3 V to ±30 V in continuous mode, the amplifier uses four MAX14808 boards in parallel connection. Each board amplifies four ±30 V output signals. The regulated DC 5 V and DC 3.3 V is supplied by P3030 power supply unit (Advantek, Hayward, CA, USA) with a maximum current of 3 A and an accuracy of ±0.1 V. The bipolar voltage from ±5 V to ±30 V is supplied by K633A power supply unit (Exso, Busan, Korea) with a maximum current of 3 A and an accuracy of ±0.1 V.

Figure 4 shows the driven signal for transducers with the phase modulation in radian at 0π, 0.05π, 0.1π, 0.15π, and the acoustic pressure in water is measured by hydrophone tip NH1000 (Precision Acoustics, Dorchester, UK).

The vision system consists of two cameras to obtain the position information of a micromotor. A Lifecam Studio Camera (Microsoft, Redmond, WA, USA) inserted into a microscope (Carl Zeiss Meditec AG, Jena, Thuringia, Germany) and a DSLR EOS 600D camera (Canon, Ota City, Tokyo, Japan) are placed on the side and front of the workspace, respectively. The total phase for each transducer can be estimated based on the feedback information from two cameras. First, the UT array traps a micromotor at the initial point (0, 0, 0) by generating a twin trap field. Then the phase of each transducer is controlled so that the initial trapping point moves to the newly updated position. One camera captures the position of the micromotor in the XZ plane and the other camera captures the position of the micromotor in the YZ plane. Based on this information, the three-dimensional current position of the micromotor is determined, and the difference between the current position and the desired position is calculated. Err X, Err Y, and Err Z represent the distance (i.e., position error) along each axis between the current position and the desired position of the micromotor. Figure 5 shows a block diagram of feedback control for three-dimensional position tracking in the water.

### 2.4. Micromotors for Drug Delivery System

The micromotors with 400-µm in diameter, which is designed for the drug delivery system, are used for the particle manipulation. Figure 6 shows the inside of the micromotor, and this part consists of a porous Poly Lactic-co-Glycolic Acid (PLGA) for drug loading [29].

## 3. Results

### 3.1. Experimental Setup

The experimental setup is shown in Figure 7. As feasibility tests for micromotors manipulation, we demonstrated 2D and 3D position control of the 400-µm micromotors using the active traveling wave from UT array with the single-side arrangement. The designed UT array is built with 16 ultrasonic transducers (JAPAN PROBE, Yokohama, JP) and placed in an acrylic box (300 × 300 × 300 ${\mathrm{mm}}^{3}$) filled with water. The custom control system generates appropriate phase delay signals for phase modulation, and the signals are transmitted to transducers to trap and manipulate the micromotor. The movement of the micromotor is captured in real time by mounted cameras for position feedback control.

The UT array needs calibration for matching the real and recognized focal point in the vision cameras. To proceed with the calibration, the real focal point needs to be defined in the first step. The hemispherical arrangement has a physical focal point at 36.47 mm from each head of the transducer, thus the maximum pressure value should be measured at the point. Using the Hydrophone (Precision Acoustics, Dorchester, UK), the z-axis of the focal point is scanned to check the pressure value, and Table 1 shows that the maximum value with 722 kPa is measured at the desired point. Based on these data, the highest-pressure level point is set as the initial focal point (0, 0, 0) in the vision program. The distance for manipulation is calculated based on this initial setup point.

### 3.2. 2D and 3D Micromotor Manipulation

We demonstrate the UT array generating active traveling wave through micromotor manipulation in 2D and 3D space. The manipulation speed can be controlled according to the displacement for single-step manipulation and its moving time. The displacement for each step manipulation should be less than the distance between two cylindrical beams in the twin trap, which is 1.5 mm on the simulation shown in Figure 2. The smallest phase delay difference between the two transducers is 25 ns, which results in the smallest displacement is 37.5 µm. The LabVIEW program on the computer sends the new signal to UT every 53 ms. Thus, the working range of the manipulation speed is from 0.707 mm/s to 28.3 mm/s. However, the smaller displacement for each step manipulation allows more accurate manipulation. Taking this into account, in the experiment, each step was determined to be 0.1 mm, where the manipulation speed was 1.89 mm/s.

In the first experiments, the 2D manipulation ability of the proposed UT array was evaluated based on the open-loop methods. The micromotor was placed in the acrylic box filled with water, and it was then manipulated to move along the “RRI” text path in XY, YZ, and ZX plane, respectively. In open-loop control, the trapping point emitted from the UT array was automatically moved along the predefined text path with 0.1 mm incremental step. Figure 8 shows the desired and tracked paths of the micromotor controlled by the open-loop control, where the maximum average error is 740.22 ± 465.23 μm in the YZ plane. The acoustic radiation forces along with *Y*- and *Z*-axis are less than that of the *X*-axis direction, as shown in the simulation results of Figure 2. Thus, manipulation ability in the YZ plane is unstable compared with other planes, furthermore, small position errors at each position in the path were accumulated, which caused the increase of the tracking position error in open-loop control.

Second, the closed-loop control method was then applied to improve the manipulation performance, where a micromotor was manipulated along the same text path. In the closed-loop control, the phase of each transducer is recalculated and emitted to minimize the position error between the desired path and the tracked position of the micromotor based on the visual feedback information. Figure 9 shows the position accuracy of closed-loop control in the 2D plane, and the micromotor could move precisely along the desired path. The maximum average error is 167.88 ± 64.59 μm in the YZ plane, which is smaller than the micromotor size.

Finally, the 3D manipulation is conducted. The array has the theoretical workspace of ‘8.66 cubic mm’, but to reduce sidelobes, ‘5 cubic mm’ is decided as the experimental workspace in 3D manipulation. The driven voltage and frequency of the transducers were kept constant. The micromotor was first trapped and moved along a rectangular path in the XZ plane and then moved to the next XZ plane with a 5 mm offset along *Y*-axis. Then finally, the micromotor was controlled again in the offset XZ plan with the same rectangular path. Figure 10 shows the experimental results of closed-loop control in 3D space, where the maximum average error is 79.77 ± 55.18 μm, 116.08 ± 65.45 μm, 108.19 ± 76.05 μm in *X*-, *Y*-, and *Z*-axis, respectively.

### 3.3. Ex Vivo Experiment

The environment inside real organs is complicated, including viscosity, non-rigid structure, and different acoustic impedance [30]. The micromotor manipulation in ex vivo condition challenges that the heterogeneous tissue decreases the permeability of each emitted ultrasonic beam. To check the permeability in the ex vivo environment, the hydrophone (Precision Acoustics, Dorchester, UK) is scanning the formed single beam, which is through 4 mm porcine fat and muscle tissue, as shown in Figure 11a. The range of -6 to +6 mm in the vertical direction is measured, and Figure 11b shows around 10 percent attenuation at the focal point. It should be verified whatever this attenuation affects particle manipulation or not, thus three-dimensional manipulation is tested under the using porcine tissue condition. The 400-μm micromotor is stably trapped at the focal point, and the micromotor is manipulated over 2 mm in the positive (or negative) direction of each axis, as shown in Figure 11c.

## 4. Conclusions

In this study, we have developed a system that is capable of generating stable 3D twin traps that can both capture and manipulate a micromotor along predefined paths through the water with position error less than 1 time of body micromotor scale. The main principles of the study include: (1) The characteristics of the proposed UT array generating active traveling waves inside the water environment are confirmed by simulation that the position of twin trap was generated by phase control of each transducer, and the radiation force of the x-axis is significantly stronger than other axes due to the feature of the trapping pattern. (2) To verify these results, micromotor manipulation tests were conducted under the open-loop system. The micromotor can draw the ‘RRI’ path in the experiment, but large errors occurred in the y and z-axis. These errors can be explained by the trapping force of these axes not being enough to control the position. (3) Thus, a 3D vision feedback control system was designed to localize and keep track of the path in real-time. By using this control system, the error is decreased in the 3D manipulation by over 60 percent.

This study will be further researched for a new target drug delivery system. Checking for the feasibility of the study, the PLGA drug micromotor was used as the main micromotor, and the performance of our system was tested in the ex vivo environment. The current vision type feedback method also will be replaced by an ultrasound imaging type for an in vivo condition.

## Figures and Tables

**Figure 1 micromachines-12-00192-f001:**
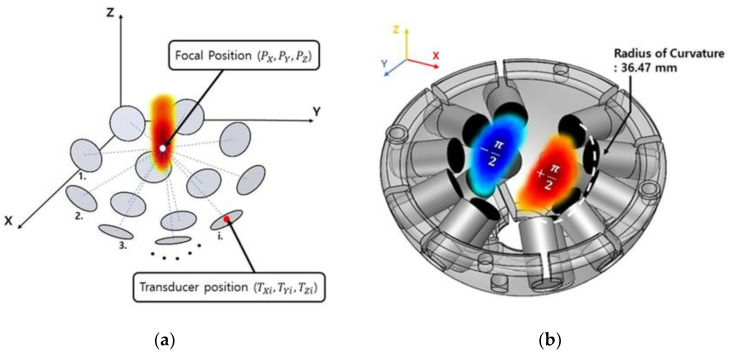
Overview of 3D manipulation: (**a**) Geometry of phased delay, (**b**) ultrasonic transducer (UT) array model with a hemispherical arrangement of transducers.

**Figure 2 micromachines-12-00192-f002:**
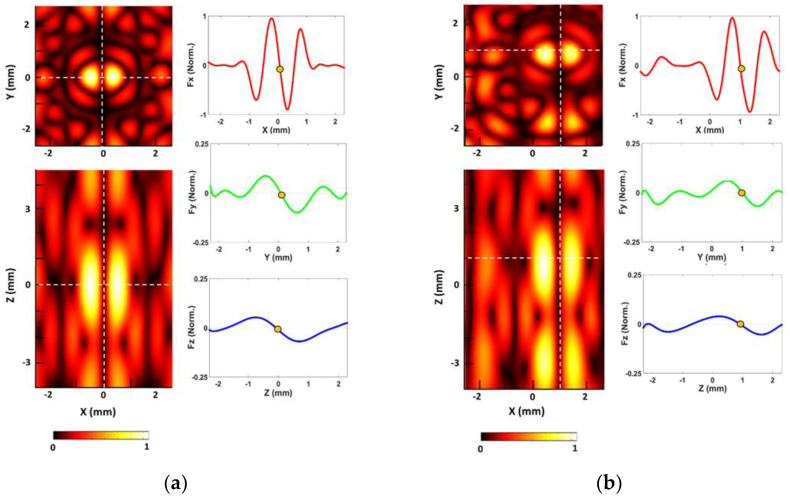
The simulated results of 3D manipulation: (**a**) Normalized acoustic radiation fields and force at the initial trapping point (0, 0, 0), and (**b**) at the changed trapping point (1, 1, 1).

**Figure 3 micromachines-12-00192-f003:**
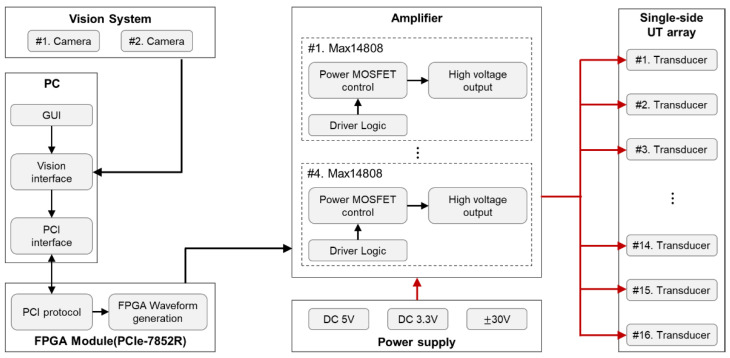
A block diagram of control hardware.

**Figure 4 micromachines-12-00192-f004:**
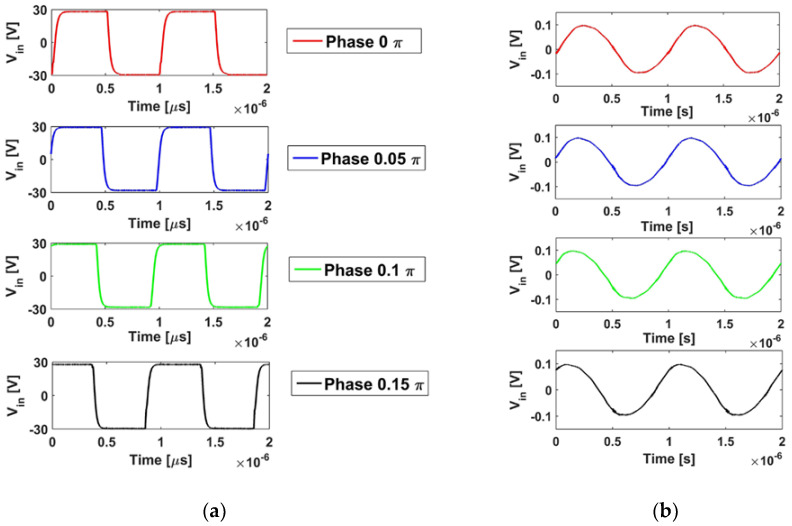
The driven voltage and measured pressure: (**a**) The phase modulation in radian of channel 1 at 0π, channel 2 at 0.05π, channel 3 at 0.1π, and channel 4 at 0.15π, (**b**) the acoustic pressure measured by hydrophone in water tank for four channels, respectively.

**Figure 5 micromachines-12-00192-f005:**
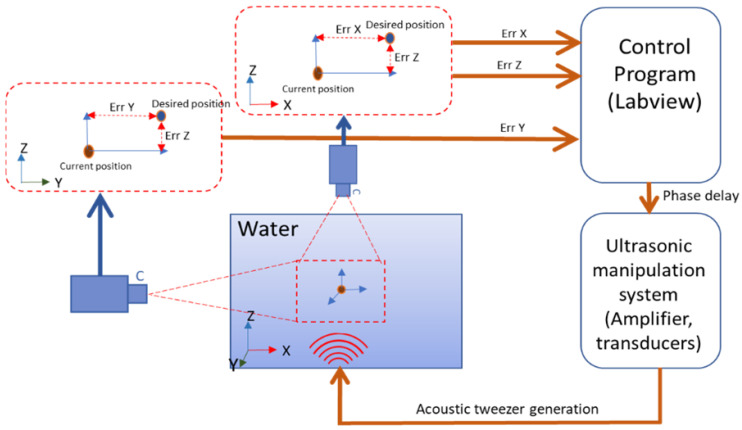
A block diagram of feedback control for three-dimensional position tracking.

**Figure 6 micromachines-12-00192-f006:**
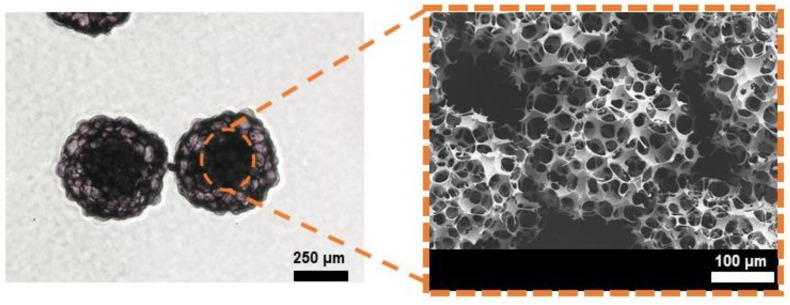
PLGA micromotors: Optical microscope image (**left**) and Scanning Electron Microscope image (**right**).

**Figure 7 micromachines-12-00192-f007:**
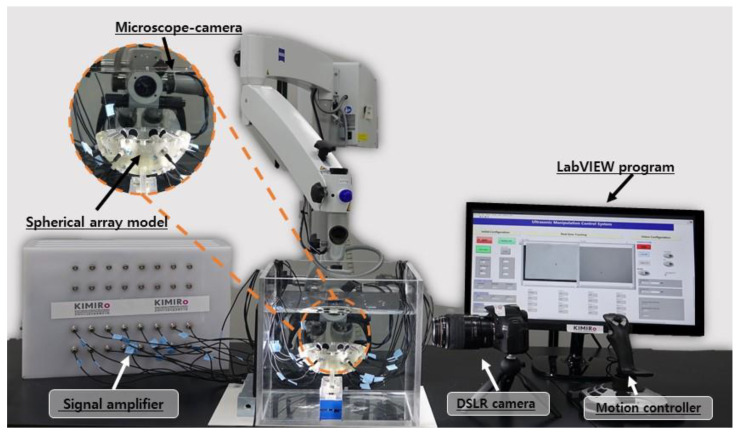
Experimental setup of ultrasonic manipulation system.

**Figure 8 micromachines-12-00192-f008:**
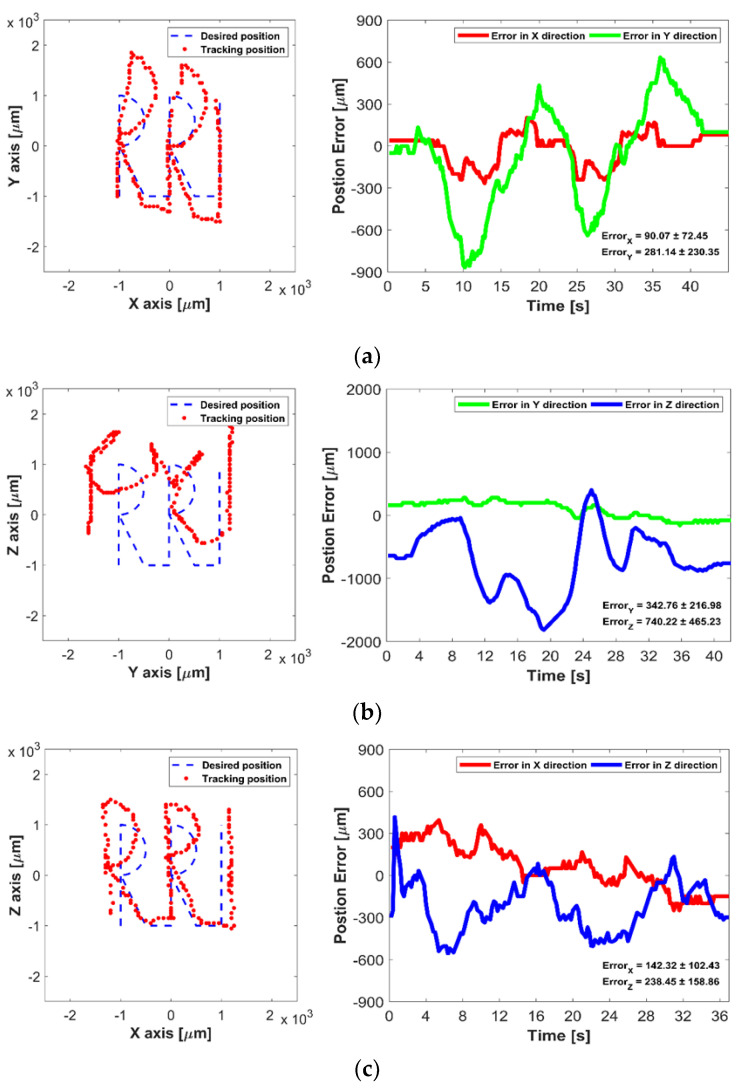
Experimental results of open-loop control in 2D plane: (**a**) Micromotor manipulation in XY plane, (**b**) in YZ plane, and (**c**) in ZX plane.

**Figure 9 micromachines-12-00192-f009:**
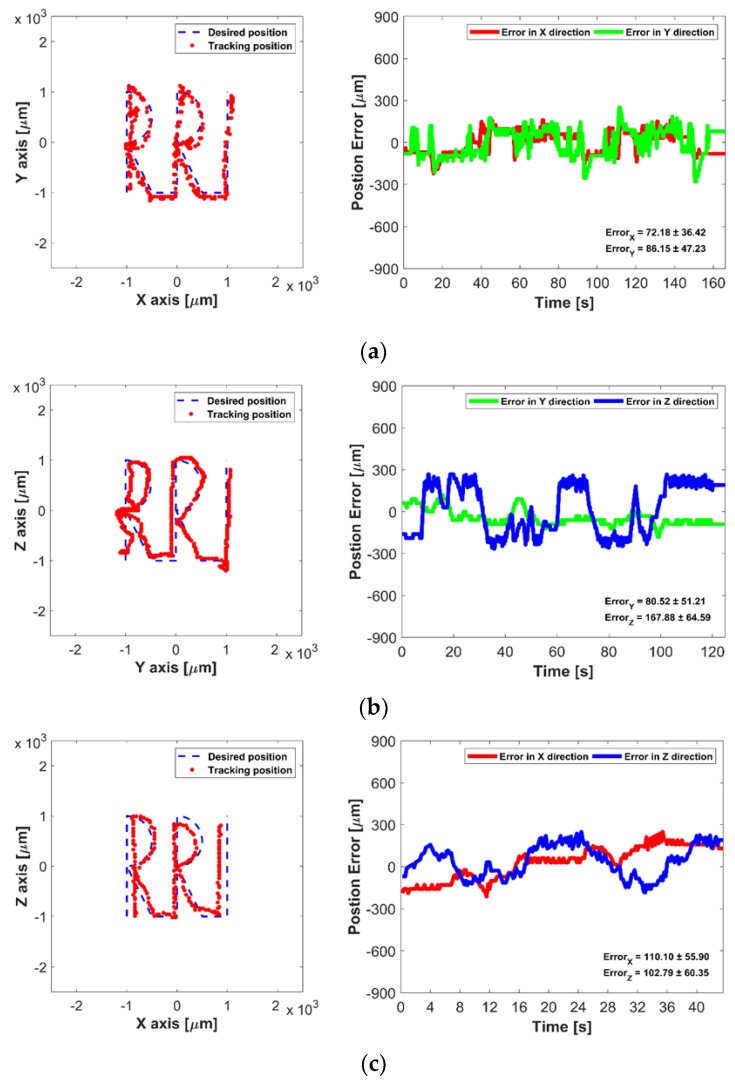
Experimental results of closed-loop control in 2D plane: (**a**) Micromotor manipulation in XY plane, (**b**) in YZ plane, and (**c**) in ZX plane.

**Figure 10 micromachines-12-00192-f010:**
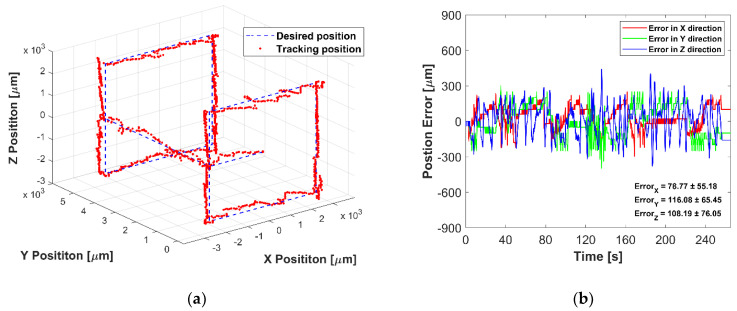
Experimental results of 3D manipulation: (**a**) The desired path and tracked position of the micromotor in 3D space, (**b**) Position error in *X*-, *Y*-, and *Z*-axis.

**Figure 11 micromachines-12-00192-f011:**
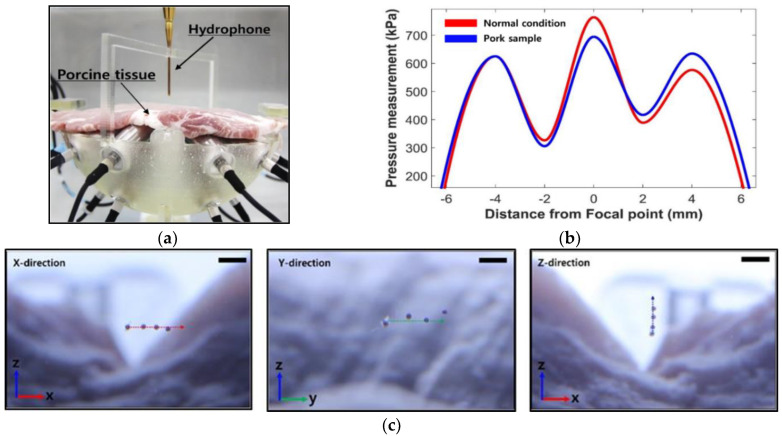
Ex vivo experiment for testing the permeability of UT array: (**a**) Experimental setup. (**b**) Result of pressure scanning at the focal point. (**c**) Time-lapse images of the manipulation in *X*-, *Y*-, and *Z* directions (Scale bar-2 mm).

**Table 1 micromachines-12-00192-t001:** Acoustic pressure measured along the Z-axis from the focal point.

Distance from the focal point [mm]	−6	−4	−2	0	2	4	6
Pressure [kPa]	269	410	194	722	139	495	108

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
