# Peer review of "Micromotor Manipulation Using Ultrasonic Active Traveling Waves"

_micromachines, 2021, doi:10.3390/mi12020192_

Round 1

Reviewer 1 Report

This paper presents an ultrasonic manipulation system which is very interesting to the research community. This work has major contributions and high quality. However, some issues need to be addressed before publication:

  1. page 4, line 119 to 129 explains how to move the focal point from 000 to 111. Please explain in more details how to achieve the backward calculation from the desired location change to the 16 transducer's phase offset.
  2. What is the workspace limit? please indicate and explain via theoretical and experimental confirmation.
  3. The pressure is presented; what about the force? It's interesting to see how heavy object can be levitated and moved.
  4. I assume the manipulation speed must have a limit. What is the working range of the manipulation speed?

Reviewer 2 Report

In the manuscript, the authors reported ultrasonic transducer (UT) array to manipulate a microcarriers in the water. The manuscript is original and novel. In my opinion, this is a good work and deserves publication. Prior to a possible publication, the following comments should be considered and the manuscript adjusted accordingly:

1)    A general explanation about microdevices (micromotors) is missing in the ‘Introduction’ section of the manuscript (what they are, how they move, application…). It would be nice to include additional clarifications so that someone who is not in this field can understand the research. 

2)    The authors should cite relevant literature regarding the use of micromotors as a drug carriers: 

  1. Maric, T., Nasir, M. Z. M., Rosli, N. F., Budanović, M., Webster, R. D., Cho, N. J., & Pumera, M. (2020). Microrobots Derived from Variety Plant Pollen Grains for Efficient Environmental Clean Up and as an Anti‐Cancer Drug Carrier. Advanced Functional Materials, 30(19), 2000112.
  2. Maric, T., Beladi‐Mousavi, S. M., Khezri, B., Sturala, J., Nasir, M. Z. M., Webster, R. D., ... & Pumera, M. (2020). Functional 2D germanene fluorescent coating of microrobots for micromachines multiplexing. Small, 16(27), 1902365.
  3. de Ávila, B. E. F., Angsantikul, P., Li, J., Lopez-Ramirez, M. A., Ramírez-Herrera, D. E., Thamphiwatana, S., ... & Wang, J. (2017). Micromotor-enabled active drug delivery for in vivo treatment of stomach infection. Nature communications, 8(1), 1-9.
  4. Medina-Sánchez, M., Xu, H., & Schmidt, O. G. (2018). Micro-and nano-motors: the new generation of drug carriers. Therapeutic delivery, 9(4), 303-316.

3) Please use ‘’micromotor’’ term instead of microcarrier as microcarriers are static microdevices.

4) Page 1, are there any other manipulations of micromotors apart from magnetic and acoustic based, and what they are?

5) The moving efficiency is related to the size of micromotors but it is not stated in the manuscript. Have you tried to manipulate the microcarriers in a different shape and different size? Would you get a similar results?

6) In page 6, a more detailed explanation regarding the structure of microcarriers should be given. How did you prepare them?

7) Can the concept apply to other microdevices in general?

8) Figures 2 should be moved to the supporting information.

9) Figures 3 should be moved to the supporting information.

10) Figures 5 should be moved to the supporting information.

11) Average velocities of microcarriers are missing.

12) The first sentence in the section 3.3 Ex-vivo experiment is incomplete and must be corrected.

Round 2

Reviewer 1 Report

My concerns have been addressed. No further questions.